# CHAIN-OF-KNOWLEDGE: GROUNDING LARGE LANGUAGE MODELS VIA DYNAMIC KNOWLEDGE ADAPTING OVER HETEROGENEOUS SOURCES

**Xingxuan Li**[1,2*†]**, Ruochen Zhao**[2*‡]**, Yew Ken Chia**[1,3*†]**, Bosheng Ding**[1,2†]**, Shafiq Joty**[2,4]
**Soujanya Poria**[3]**, Lidong Bing**[1,5]
[1]DAMO Academy, Alibaba Group, Singapore, [2]Nanyang Technological University,
[3]Singapore University of Technology and Design, [4]Salesforce Research,
[5]Hupan Lab, 310023, Hangzhou, China
{xingxuan.li, yewken.chia, bosheng.ding, l.bing}@alibaba-inc.com
{ruochen002, srjoty}@ntu.edu.sg   sporia@sutd.edu.sg

## ABSTRACT

We present chain-of-knowledge (CoK) , a novel framework that augments large language models (LLMs) by dynamically incorporating grounding information from heterogeneous sources. It results in more factual rationales and reduced hallucination in generation. Specifically, CoK consists of three stages: reasoning preparation, dynamic knowledge adapting, and answer consolidation. Given a knowledge-intensive question, CoK first prepares several preliminary rationales and answers while identifying the relevant knowledge domains. If there is no majority consensus among the answers from samples, CoK corrects the rationales step by step by adapting knowledge from the identified domains. These corrected rationales can plausibly serve as a better foundation for the final answer consolidation. Unlike prior studies that primarily use unstructured data, CoK also leverages structured knowledge sources such as Wikidata and tables that provide more reliable factual information. To access both unstructured and structured knowledge sources in the dynamic knowledge adapting stage, we propose an adaptive query generator that allows the generation of queries for various types of query languages, including SPARQL, SQL, and natural sentences. Moreover, to minimize error propagation between rationales, CoK corrects the rationales progressively using preceding corrected rationales to generate and correct subsequent rationales. Extensive experiments show that CoK consistently improves the performance of LLMs on knowledge-intensive tasks across different domains. Our code is available at https://github.com/DAMO-NLP-SG/chain-of-knowledge.

## 1 INTRODUCTION

In recent years, large language models (LLMs) such as ChatGPT (OpenAI, 2023) have demonstrated impressive language generation capabilities (Cheng et al., 2023; Ding et al., 2023; Chen et al., 2024). However, one major challenge of LLMs lies in hallucination, which is their tendency to confidently generate plausible but factually incorrect texts (Ji et al., 2023; Zhao et al., 2023b). As shown in Figure 1, given a question, "What year was the Argentine actor who directed El Tio Disparate born?" which requires factual knowledge to answer, the most advanced LLMs often provide an incorrect answer. While LLMs have the remarkable capability to recall information from their training data, effectively updating or controlling the factual knowledge within these models remains challenging (Luo et al., 2023).

---

* Equal contribution.
† Xingxuan Li, Yew Ken Chia, and Bosheng Ding are under the Joint Ph.D. Program between DAMO Academy and their corresponding universities.
‡ Ruochen Zhao is under the AISG Ph.D. Fellowship Programme.

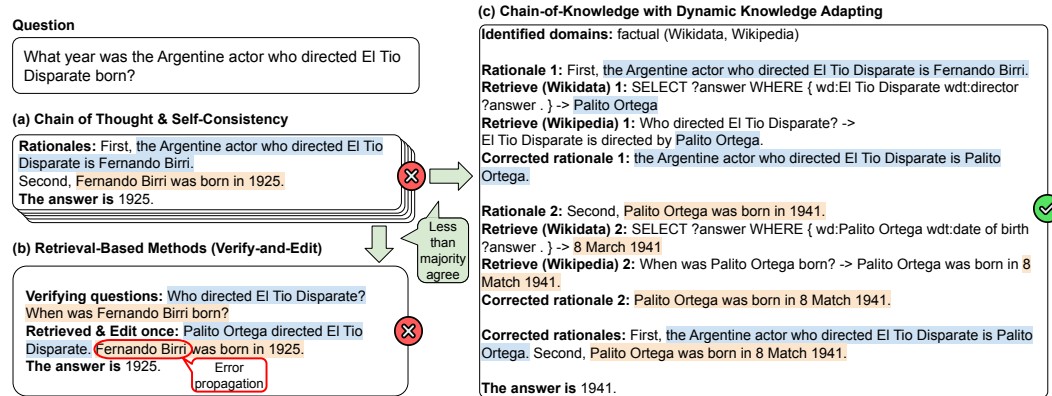

Figure 1: Comparison of different methods: (a) chain-of-thought with self-consistency (Wei et al., 2022), (b) verify-and-edit (Zhao et al., 2023c), and (c) chain-of-knowledge or CoK (this work). CoK incorporates heterogeneous sources for knowledge retrieval and performs dynamic knowledge adapting. For clarity and succinct presentation, only pivotal steps are shown in the figure. Refer to Appendix A for the prompt design of each method.

A promising direction to address hallucination in generation is to augment the LLMs with external knowledge (Mialon et al., 2023). These methods involve incorporating LLMs with a retrieval system, which seeks to utilize external factual knowledge to guide the generation process. Instead of relying solely on the internal training knowledge of LLMs, these methods can fetch relevant information from external knowledge sources, such as web documents (Shi et al., 2023) and knowledge bases (Xie et al., 2022). Furthermore, to tackle more complex questions that require intricate reasoning, Zhao et al. (2023c) recently proposed a Verify-and-Edit (VE) framework, which improves chain-of-thought (CoT) reasoning (Wei et al., 2022) of LLMs by incorporating a retrieval system.

However, these methods have three inherent limitations. First, they use a fixed knowledge source for all questions, which may fail to retrieve specialized and domain-specific knowledge. For instance, it may not be effective to query Wikipedia for a medical question. Second, to generate retrieval queries, existing methods primarily rely on LLMs, which are predominantly pre-trained on natural language sentences, and thus may not be effective in generating structured queries like SPARQL, which is used to query knowledge graphs. Third, existing retrieval-augmented methods lack progressive correction capability, leading to potential error propagation. For example, in Figure 1, we define each rationale to be a thought step (sentence) within the CoT. Verify-and-Edit retrieves knowledge for each rationale in parallel and independently. Since the second rationale depends on the first, errors can carry over from the *verification* step to the *edit* step, making the retrieved knowledge misaligned with each other and the actual question, resulting in an incorrect final answer. Similarly, ReAct (Yao et al., 2023) also leaves errors from prior (*reason* or *act*) steps in the prompt, causing potential noise and bias for LLM inference.

To address these limitations, we propose chain-of-knowledge (CoK), a framework that augments LLMs dynamically using heterogeneous knowledge sources. An example of how CoK functions is shown in Figure 1, for the question, "What year was the Argentine actor who directed El Tio Disparate born?", CoT with self-consistency (Wei et al., 2022) is first utilized to generate preliminary rationales, pinpoint the relevant knowledge domains, and select answers that lack a majority consensus for further processing. In the subsequent dynamic knowledge adapting stage, an adaptive query generator (AQG) is employed to generate queries for the knowledge sources within the selected domains. To effectively retrieve knowledge with heterogeneous formats, AQG can adaptively generate queries of the corresponding types, such as SPARQL and natural sentence (see Figure 2). Subsequently, by executing the generated queries, supporting knowledge is obtained and utilized to edit the first rationale (*i.e.,* rectify the director from Fernando Birri to Palito Ortega); it ensures mistakes do not propagate into the subsequent generation of the second rationale. The same process is then applied to edit the second rationale. Finally, with the corrected chain of rationales, CoK derives the final answer.

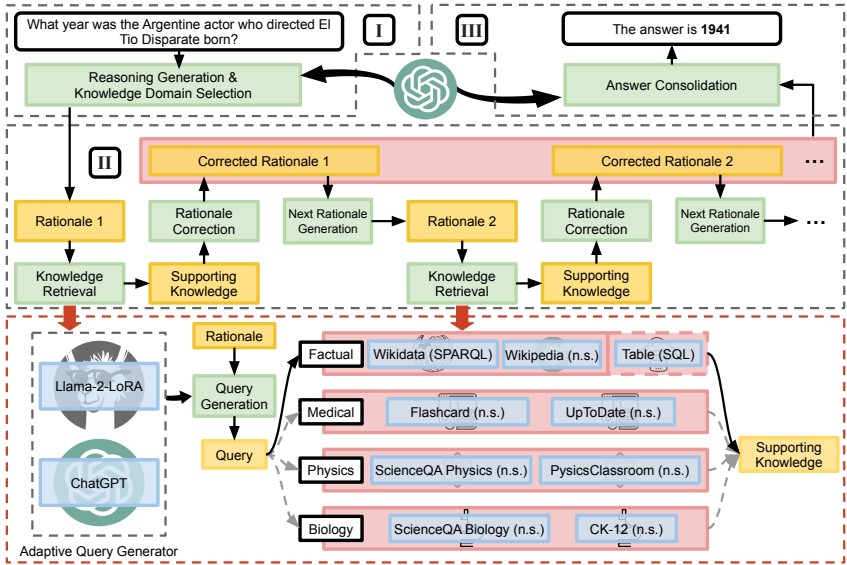

Figure 2: Our proposed chain-of-knowledge (CoK) framework, consisting of (I) Reasoning preparation, (II) Dynamic knowledge adapting, and (III) Answer consolidation. n.s.: natural sentence.

Given that different knowledge sources require distinct query languages, AQG holds a crucial role in generating queries. AQG is versatile and can either be a fine-tuned model like Llama-2 (Touvron et al., 2023) with LoRA (Hu et al., 2021) or an off-the-shelf LLM like ChatGPT. By leveraging both unstructured and structured knowledge sources, CoK allows for better factual accuracy, improved reliability, and easier information updates.

To summarize, our key contributions are the following: (1) We introduce chain-of-knowledge (CoK), a novel framework to enhance the factual correctness of LLMs with heterogeneous knowledge sources; (2) We propose an adaptive query generator (AQG), specially designed to generate queries tailored to each knowledge source. AQG is versatile and can seamlessly transition between fine-tuned models and black-box LLMs; (3) CoK corrects the rationales progressively, ensuring that inaccuracies from preceding rationales do not propagate into the subsequent steps; (4) We perform extensive experiments on knowledge-intensive tasks spanning a range of domains, including factual, medical, physical, and biological. CoK outperforms the CoT baseline by 4.3% on average.

## 2 THE CHAIN-OF-KNOWLEDGE FRAMEWORK

As shown in Figure 2, the CoK framework consists of three stages: (1) reasoning preparation, (2) dynamic knowledge adapting, and (3) answer consolidation. In the first stage, given a knowledge-intensive question, CoK generates preliminary rationales, *i.e.,* reasoning units/sentences in the reasoning chain of CoT, and answers while identifying the relevant knowledge domains. Questions that do not yield a majority consensus in their answers enter the dynamic knowledge adapting stage, in which an adaptive query generator (AQG) is employed to generate queries to retrieve knowledge from the knowledge sources of the identified domain. The rationales are progressively revised and generated based on the retrieved knowledge. The final answer is then derived based on the corrected rationales. Refer to Appendix A.1 for the prompts used for each step of our framework.

### 2.1 REASONING PREPARATION STAGE

In real-world scenarios, when facing a complex knowledge-intensive question, it is necessary to generate intermediate rationales before producing the final answer (Wei et al., 2022). Moreover, before delving into external knowledge sources to address the question, it is crucial to identify the relevant knowledge domains for effective retrieval. Thus, the reasoning preparation stage consists of two essential components, namely, reasoning generation and knowledge domain selection.

Table 1: An example of generated query, execution results, and formatted knowledge for rationales of SPARQL. Knowl. stands for knowledge.

| SPARQL | |
|---|---|
| **Rationale** | Souleyman Sané's son, Leroy Sané, is a professional football player. |
| **Generated query** | SELECT ?answer WHERE { wd:/Souleymane Sané/ wdt:/child/ ?answer . } |
| **Execution results** | Leroy Sané |
| **Formatted knowl.** | The fact entity of the sentence "Souleyman Sané's son, Leroy Sané, is a professional football player" is Leroy Sané. |

**Reasoning Generation** Previous studies have demonstrated the importance of intermediate rationales for LLMs to answer complex reasoning questions (Wei et al., 2022). In this work, we utilize the few-shot chain-of-thought (CoT) prompting to generate rationales (Wei et al., 2022). Moreover, we employ the self-consistency method (Wang et al., 2023) to determine whether external knowledge is necessary to answer the question. In sampling various reasoning paths and answers, self-consistency is found to be highly correlated with accuracy. Thus, predictions with high consistency are preserved without modification. Only questions with "uncertain" answers, *i.e.,* their consistency falls below a specified threshold, undergo further stages of processing. Such filtering technique is found to be useful in identifying incorrect predictions by previous works (Yao et al., 2023; Zhao et al., 2023c).

**Knowledge Domain Selection** To ensure the retrieval of the most pertinent knowledge to the question, we introduce the knowledge domain selection step. As shown in Figure 2, CoK integrates four distinct knowledge domains: factual, medical, physics, and biology. Moreover, multiple domains can be identified for answering a single question. To illustrate, when presented with the question "Who proposed the theory which explains the cause of tides?", both physics (gravitational force of the Moon causes tides) and factual (Isaac Newton first proposed the universal gravitation and explained tidal forces exerted by celestial bodies) domain knowledge are required to answer the question. The knowledge domain selection is completed through in-context learning.

## 2.2 DYNAMIC KNOWLEDGE ADAPTING STAGE

Once the preliminary rationales and the identified knowledge domains are obtained, the next stage is dynamic knowledge adapting, *i.e.,* rectifying rationales based on the retrieved knowledge. To minimize error propagation, CoK conducts knowledge retrieval and correction of the rationales sequentially. The preceding corrected rationales are used to generate the next rationale, which then undergoes the same knowledge retrieval and correction step.

**Knowledge Retrieval** Upon identifying relevant domains to the question in the reasoning preparation stage, all knowledge sources within these domains are utilized for knowledge retrieval. The knowledge retrieval consists of two steps: query generation and execution.

**A) Query Generation** Depending on the nature of the knowledge sources, each source is linked to the most appropriate query language, which could either be structured, such as SPARQL or SQL, or unstructured, such as natural language sentences. For instance, Wikidata is linked to the SPARQL query as it consists of knowledge graphs. The flashcard source is linked to the natural sentence query as it takes the format of natural sentence pairs. An example of generated queries for SPARQL is shown in Table 1. [1] For instance, given a sentence "Souleyman Sané's son, Leroy Sané, is a professional football player", a SPARQL query, "`SELECT ?answer WHERE {wd:/Souleymane Sané/ wdt:/child/ ?answer.}`", is generated to retrieve relevant knowledge from Wikidata. To facilitate the generation of both structured and unstructured queries, an adaptive query generator (AQG) is used. AQG is a versatile plug-in component, which can be either a tailor-finetuned model or an off-the-shelf LLM. Details of AQG will be elaborated in Section 3.

**B) Query Execution** Once the queries are generated, the subsequent step is their execution to acquire and convert the knowledge into formatted knowledge (see Table 1). A specialized method is devised to execute queries and format the results for each query language. For SPARQL queries, entity linking is initially performed to substitute entity spans with IDs, followed by acquiring results by invoking the API of `wikidata.org`. Regarding SQL queries, they are executed directly to

---

[1]Examples of generated queries for each querying language are in Appendix D.5.

fetch the results, which could be a singular value or a subset of the original table. The outcomes from both SPARQL and SQL are then formatted into markdown text. For natural sentence queries, knowledge is retrieved from domain-specific knowledge sources either through sentence similarity matching or by utilizing a search engine. [2]

**Rationale Correction**  Existing methods such as ReAct (Yao et al., 2023) and Verify-and-Edit (Zhao et al., 2023c) keep all retrieved information in the context throughout the process, no matter if it contains reasoning mistakes. This often leads to error propagation and misguides further generations. To overcome this weakness, CoK involves a progressive rationale correction step. Given the current rationale and the formatted knowledge from various knowledge sources, a corrected rationale is generated to replace the current one. This step helps in rectifying any factual incorrectness and preventing error propagation.

**Next Rationale Generation**  Using the question and preceding corrected rationales, the next rationale is generated, and the process is reiterated for the new rationale until a final answer is produced.

## 2.3   Answer Consolidation Stage

Ultimately, the LLM is prompted with the question and corrected rationales to generate a consolidated answer, which is expected leading to a more accurate answer. This hypothesis is further examined through a series of experiments, as detailed in Section 4.

## 3   The Adaptive Query Generator

CoK incorporates heterogeneous knowledge sources from four different domains, including factual, medical, physics, and biology. Each of these knowledge sources necessitates the use of a unique query language for retrieval, which could be either structured or unstructured. Therefore, we design the adaptive query generator (AQG) to facilitate query generation for different knowledge sources.

**Unstructured Query Languages**  Natural language sentences are the most natural way that human beings search for information. AQG utilizes two distinct approaches for generating unstructured queries based on the knowledge sources. **A)** For general factual knowledge sources, such as Wikipedia, ChatGPT is utilized. **B)** For domain-specific knowledge sources (*e.g.,* Flashcard, ScienceQA Physics, and ScienceQA Biology), using ChatGPT may lead to hallucination as it may not have comprehensive knowledge of the specific domains. Therefore, we instruction-tune LLaMA-2-7B using LoRA with pairs of input texts and output queries. Furthermore, the domain of the training data is on par with the respective knowledge source. Consequently, the AQG is equipped with the requisite knowledge for generating queries with greater precision.

**Structured Query Languages**  Querying unstructured knowledge sources often leads to the retrieval of irrelevant and redundant information. On the other hand, structured knowledge sources (*e.g.,* Wikidata and tables) provide direct factual results. To generate structured queries, AQG utilizes two approaches based on the query languages. **A)** When generating commonly used query languages like SQL, we employ ChatGPT. It is empirically inferred that ChatGPT included SQL during its pre-training, providing it with advantages in generating SQL queries (OpenAI, 2023). All pertinent details are incorporated into the prompt to enhance the precision of query generation. For instance, when generating SQL queries, we include both the table schema and data snippets. **B)** For less common languages like SPARQL, we instruction-tune LLaMA-2-7B using LoRA with sentence-SPARQL pairs. The training data is collected to match the logical granularity of the rationales, thereby facilitating more accurate query generation. For example in SPARQL, both training data and rationales contain single entity and relation within each sentence. Inspired by chain-of-hindsight (Liu et al., 2023), besides giving the correct queries, we also append negative examples such as "incorrect queries:.." during instruction-tuning.

Detailed query language, model, and training datasets of each knowledge source are in Table 8 of Appendix. The constructions of instruction-tuning datasets and training details are in Appendix D. We also evaluate the performances of AQG in Appendix F.2.

---

[2]Details of the execution process for each knowledge source is in Appendix C.

Table 2: Main experimental results across various domains. Acc.: accuracy. E.M.: exact match.

| Method | Factual | | | Medical | Physics | Biology |
|---|---|---|---|---|---|---|
| | **FEVER** Acc. | **HotpotQA** E.M. | **FeTaQA** BLEU | **MedMCQA** Acc. | **MMLU Physics** Acc. | **MMLU Biology** Acc. |
| Standard (3-shot) | 51.8% | 22.7% | 20.7 | 61.6% | 44.3% | 80.6% |
| CoT (3-shot) | 57.8% | 29.9% | 17.3 | 59.6% | 41.9% | 81.5% |
| CoT-SC (3-shot) | 59.9% | 30.8% | - | 60.3% | 42.7% | 81.1% |
| VE (3-shot) | 60.6% | 31.8% | 21.6 | 67.8% | 39.9% | 81.9% |
| CoK (3-shot) | **63.4%** | **34.1%** | **25.0** | **70.5%** | **45.5%** | **83.0%** |
| Standard (6-shot) | 53.4% | 24.0% | 23.1 | 64.4% | 44.7% | 81.1% |
| CoT (6-shot) | 55.6% | 34.4% | 19.4 | 66.4% | 43.5% | 81.7% |
| CoT-SC (6-shot) | 56.2% | 33.4% | - | 65.8% | 42.7% | 82.2% |
| VE (6-shot) | 57.2% | 34.4% | 23.1 | 67.1% | 43.1% | 78.9% |
| CoK (6-shot) | **58.5%** | **35.4%** | **26.0** | **73.3%** | **47.0%** | **84.4%** |

## 4 EXPERIMENTS

### 4.1 SETUP

**Models** In our experiments, we utilize ChatGPT (`gpt-3.5-turbo-0613`) as the black-box LLM for the reasoning preparation and answer consolidation stages. To ensure reproducibility, we fixed the decoding temperature to 0 for all generations. Except for the self-consistency step, we set the temperature to 0.7, allowing for the sampling of five rationales and answers, as recommended by Wang et al. (2023). When less than half of the answers agree [3], we edit the results with CoK.

**Knowledge Sources** We choose authoritative knowledge sources for each domain. Specifically, for the factual domain, we use Wikidata, Wikipedia, and Wikitables; for the medical domain, we use medical Flashcard and UpToDate; for physics, we refer to ScienceQA Physics and PhysicsClassroom; and for biology, we utilize ScienceQA Biology and CK-12. Details are in Appendix C, D.

**Tasks** We collect a set of knowledge-intensive tasks from various domains, including FEVER (Thorne et al., 2018), HotpotQA (Yang et al., 2018), and FeTaQA (Nan et al., 2022) in the factual domain; MedMCQA (Pal et al., 2022) in the medical domain; Physics and Biology tests from MMLU (Hendrycks et al., 2021) in the physics and biology domains. Details are in Appendix E.

**Baselines** We compare CoK with both widely used baselines and state-of-the-art methods to provide a more comprehensive overview: A) Standard prompting (**Standard**) directly predicts the answer (Ouyang et al., 2022). B) Chain-of-thought (**CoT**) (Wei et al., 2022) generates several intermediate rationales before the final answer to improve the complex reasoning capability of LLMs. C) CoT with self-consistency (**CoT-SC**) (Wang et al., 2023) replaces the naive greedy decoding in CoT with sampling a diverse set of rationales and outputs the most consistent [4] answers. D) Verify-and-Edit (**VE**) (Zhao et al., 2023c) is a state-of-the-art, CoT-based framework that seeks to improve the prediction factuality by post-editing rationales with external knowledge. E) **ReAct** (Yao et al., 2023) combines agent thoughts and open-domain knowledge search to reach a final answer. [5] Following the baselines, we evaluate using the few-shot setting and ensure that all methods use the same number of demonstration samples. [6]

### 4.2 RESULTS

**CoK Consistently Outperforms CoT** As shown in Table 2, CoK consistently outperforms CoT and CoT-SC on each dataset. On factual-domain tasks, the average improvement on 3-shot and

---

[3] On FEVER, as the output space is limited to 3 answer choices, we always have high consistency. Thus, we use less than 4 out of 5 of the answers agree.

[4] Note that self-consistency is not applicable for FeTaQA as it is an open-ended generation task, and we can have near-equivalent generations that are nevertheless not exact matches. More details are in Appendix B.

[5] We report the results for ReAct separately in Table 3 as it uses the PaLM model (Chowdhery et al., 2022).

[6] Detailed prompts for baselines are in Appendix A. Our work focuses on few-shot settings, thus not including supervised methods as baselines.

Table 3: Results of retrieval-based methods on FEVER and HotpotQA. ReAct results are adapted from Yao et al. (2023).

| Method | FEVER (3-shot) | | HotpotQA (6-shot) | |
|---|---|---|---|---|
| | Acc. | Δ Acc. | E.M. | ΔE.M. |
| CoT-SC→ReAct | **64.6%** | **+4.2%** | 34.2% | +0.8% |
| ReAct→CoT-SC | 62.0% | +1.6% | 35.1% | +1.7% |
| CoT-SC | 59.9% | - | 33.4% | - |
| Verify-and-Edit | 60.6% | +0.7% | 34.4% | +1.0% |
| CoK (ours) | 63.4% | +3.5% | **35.4%** | **+2.0%** |

Table 4: Results of using single or multiple knowledge domains and sources on MedMCQA (3-shot).

| Method | Knowl. Domains | Knowl. Sources | Acc. |
|---|---|---|---|
| CoT | - | - | 59.6% |
| CoK | Medical | Flashcard | 67.1% |
| CoK | Medical | Flashcard, UpToDate | 69.2% |
| CoK | Medical, Biology | Flashcard, UpToDate, ScienceQA, CK-12 | 70.5% |

6-shot is prominent on HotpotQA and FEVER, registering at 2.6% and 4.3% respectively. This suggests that CoK is not only effective on multi-step reasoning datasets (HotpotQA), but benefits less single-hop datasets (FEVER) as well with its accurate retrieval abilities. On domain-specific datasets, such as MedMCQA, and MMLU Physics and Biology, CoK achieves an average accuracy improvement of 4.9% over the CoT baseline on 3-shot and 6-shot settings. We notice that CoT has worse performances than standard prompting on FetaQA, MedMCQA, and MMLU Physics. This illustrates that, while CoT is effective for addressing complex reasoning tasks, it struggles with hallucination in its rationales when handling knowledge-intensive tasks, leading to incorrect answers. This outcome aligns with the findings of Yao et al. (2023) and Zhao et al. (2023c) as well. With dynamic knowledge adapting, CoK can effectively reduce hallucination in the rationales and we include analysis on the factual accuracy in Section 5.3.

**CoK vs. Other Retrieval-based Methods** As shown in Table 2, CoK consistently outperforms state-of-the-art retrieval-based method Verify-and-Edit (VE) (Zhao et al., 2023c). For FEVER and HotpotQA, we additionally compare with the results reported in ReAct (Yao et al., 2023) in Table 3. Since the results in ReAct are reported on the PaLM model (Chowdhery et al., 2022), to add a more justified perspective, we report the performance improvement gained on top of the CoT-SC baseline. Compared with ReAct, CoK demonstrates a more substantial improvement over CoT-SC, especially on HotpotQA. More specifically, for HotpotQA, CoK exhibits improvements of 2.0% compared to 0.8% by ReAct. On FEVER, CoK shows a 3.5% improvement, which is on par with the 4.2% improvement gained by ReAct. This is attributed to the fact that FEVER is less multi-hop compared to HotpotQA, thus benefitting less from an improved CoT. VE conducts knowledge retrieval and editing for all rationales in parallel, and ReAct leaves past errors in the prompt, potentially leading to error propagation. CoK alleviates this issue with progressive knowledge adapting. It is also worth noting that CoK costs much less than ReAct, shown with a detailed cost analysis in Appendix I

**Effect of Number of Demonstrations** As shown in Table 2, CoK consistently exhibits enhanced performance across multiple datasets under both 3-shot and 6-shot settings. Several studies show that increasing the number of demonstrations (shots) in the prompt can potentially lead to better performances on reasoning tasks (Wei et al., 2022). However, this is not universally true for knowledge-intensive tasks. For example, as shown in Table 2, the performance of CoT on MMLU Biology with six shots (81.7%) is nearly identical to that with three shots (81.5%). This occurs because the bottleneck for LLMs in answering knowledge-intensive questions accurately is their insufficient knowledge, not their reasoning capability. The performance on FEVER for all reasoning-based methods decrease with six shots. This is likely due to the fact that FEVER questions are single-hop and require less reasoning. Thus, increased guidance on reasoning could

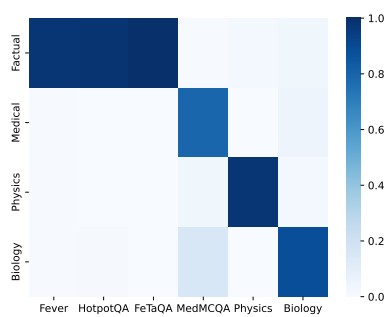

Figure 3: A heatmap on distributions of identified domains of each dataset.

lead to potential noise. This finding is consistent with ReAct (Yao et al., 2023), where the authors state that increasing beyond 3-shot for FEVER does not lead to better performance.

Table 5: Parallel vs. dynamic knowledge adapting.

| Method | HotpotQA (3-shot) |
|---|---|
| CoT | 29.9% |
| Verify-and-Edit | 31.8% |
| CoK (parallel) | 31.2% |
| CoK (dynamic) | 34.1% |

Table 6: Comparison of the factual accuracy of rationales on HotpotQA.

| Method | Rationale 1 | Rationale 2 |
|---|---|---|
| CoT-SC | 54.3% | 52.1% |
| CoK | 66.3% | 69.5% |

Table 7: Human study results on the factuality of reasoning chains.

| Predictions | CoK | CoT-SC | Tie |
|---|---|---|---|
| Correct predictions | **68%** | 4% | 28% |
| Incorrect predictions | **44%** | 24% | 32% |
| All predictions | **56%** | 14% | 30% |

## 5 ANALYSIS

### 5.1 SINGLE VS. MULTIPLE KNOWLEDGE DOMAINS AND SOURCES

As outlined in Section 2.1, CoK integrates a step to select the appropriate knowledge domains for each question. This step is crucial to ensure that CoK can retrieve the most pertinent knowledge to correct the rationales and answer the questions accurately. It is possible that multiple knowledge domains can be chosen for one question, and within each domain, there are several knowledge sources. In this subsection, we investigate the necessity of utilizing multiple knowledge domains and sources. We also include an evaluation of the domain selection performance in Appendix F.1.

**Single vs. Multiple Knowledge Domains**  We show the domain distributions identified for each dataset in Figure 3. Notably, we find that CoK predominantly selects one knowledge domain for each dataset, while a small number of cases call for multiple domains. For instance, the primary knowledge domain for MedMCQA is Medical, and 17.8% of the questions identify Biology as a relevant domain as well. Furthermore, we conduct ablation experiments to demonstrate the necessity of utilizing multiple domains. As shown in Table 4, compared to only using Medical domain knowledge, CoK using additional knowledge from the Biology domain further improves the performance by 1.3%. This indicates that knowledge spanning multiple domains is needed for answering some questions, underscoring the necessity of incorporating various knowledge domains.

**Single vs. Multiple Knowledge Sources**  Within one domain, numerous credible knowledge sources exist, and it is unfeasible for a single source to encompass all knowledge from the domain. Therefore, it is important to utilize multiple knowledge sources within one domain. For instance, as shown in Table 4, the performance of CoK improves by 2.1% when utilizing both Flashcard and UpToDate as medical knowledge sources, compared to using only Flashcard. [7]

### 5.2 PARALLEL VS. DYNAMIC KNOWLEDGE ADAPTING

As aforementioned, dynamic knowledge adapting helps CoK prevent error propagation, here we take a closer look at how much improvement it brings in. As shown in Table 5, the performance of CoK improves by 4.2% compared with CoT when dynamic knowledge adapting is applied. However, parallel editing leads to poorer performance due to error propagation for rationales.

### 5.3 EVALUATING FACTUALITY IMPROVEMENT OF THE RATIONALES

While the main results have shown that CoK effectively improves the performance of LLMs in knowledge-intensive tasks, we are also interested in reducing hallucination for the generated rationales. Hence, we conduct quantitative and qualitative evaluations to assess the factual accuracy.

**Quantitative Evaluation**  To automatically evaluate how CoK can reduce hallucination in the model outputs, we employ an existing fact-checking method to compare the original and corrected rationales. Specifically, we use ProgramFC (Pan et al., 2023) which is a state-of-the-art method for judging the factuality of claims with respect to Wikipedia. As shown in Table 6, we observe that CoK has improved factual accuracy compared to the CoT-SC baseline on the HotpotQA dataset. Notably, the factual accuracy of CoT-SC decreases for rationale 2 compared to rationale 1, which could

---

[7]Note that using external sources may have limitations such as noise or conflicts between different sources. We mainly address this by selecting authoritative knowledge sources, and discuss this further in Appendix G

be due to error propagation. On the other hand, the factual accuracy of CoK improves slightly for the second rationale, which indicates that correcting previous rationales helps the LLM to generate more factual rationales in future steps.

**Human Evaluation**    To qualitatively examine whether CoK could output factually consistent reasoning chains, we also conducted a human study. Specifically, two volunteers are given 100 outputs randomly selected from HotpotQA and FEVER datasets. The selected outputs are balanced, where 50 CoK outputs resulted in incorrect answers, and the other 50 resulted in correct answers. The volunteers are asked to select which reasoning chain is factually correct, or if there is a tie. Then, they are asked to answer whether the better CoT should lead to better results. Details on the instructions and setup can be found in Appendix H.1. The results are given in Table 7. We could observe that volunteers consistently confirm that CoK-generated reasoning chains are factually consistent while the CoT-SC chains are not. For incorrect predictions, humans still believe that 44% of the time, the CoK-generated CoT is improved on factual consistency, although it may not contain the necessary information for a correct answer. Among these instances, humans believe 73% of the time that these improved CoTs should have led to better answers. This implies that, even though the CoT quality has been improved, many failure cases are caused by reasoning errors. Case studies can be found in Appendix H.2. In general, the two volunteers show a Cohen Kappa's agreement of 0.43, which is considered moderate agreement (Landis & Koch, 1977).

## 6    RELATED WORK

**Knowledge-Intensive NLP**    While language models can generate highly coherent text and demonstrate reasoning abilities, many real-world tasks require knowledge beyond the local context. For example, fact-checking tasks may require models to locate suitable evidence on the internet or refer to external knowledge (Thorne et al., 2018). In the realm of natural language processing (NLP), a task is deemed to be knowledge-intensive when it exceeds the reasonable expectation of human capability to solve it without access to external knowledge. The resolution of such knowledge-intensive NLP tasks typically involves the utilization of retriever-reader systems. Initially, a retriever extracts a limited collection of pertinent documents from the knowledge source, after which a reader employs the context extracted to generate an appropriate response (Chen et al., 2017; Lewis et al., 2020; Guu et al., 2020). Hence, there is an urgent need to develop effective models for knowledge-intensive tasks (Petroni et al., 2021).

**Augmented Language Models**    The discipline of augmented language models (ALMs) addresses hallucinations of traditional LLMs by equipping them with improved reasoning capabilities and the capacity to utilize external resources (Chung et al., 2022). Furthermore, LLMs can learn to leverage external tools or models to accomplish the relevant tasks (Schick et al., 2023; Shen et al., 2023). ALMs can employ these enhancements independently or combine them in a specific order to complete a given task, ultimately resulting in enhanced capabilities (Mialon et al., 2023; Zhao et al., 2023a). However, previous works do not consider knowledge from multiple domains and lack progressive editing throughout the generation process, which could lead to error propagation. In this work, we propose an efficient framework to solve knowledge-intensive tasks by progressively augmenting them with diverse sources of external knowledge.

## 7    CONCLUSIONS

In this paper, we introduce chain-of-knowledge (CoK), a novel framework designed to enhance the factual correctness of large language models (LLMs). CoK represents a promising and comprehensive solution to progressive knowledge-grounded generation by incorporating heterogeneous sources in multiple domains. We address the challenge of accurate query generation by proposing the adaptive query generator (AQG) which supports both unstructured and structured query languages. The AQG can be easily transitioned between fine-tuned models and black-box LLMs. Our experimental results on knowledge-intensive tasks demonstrate the substantial improvement achieved by CoK. Furthermore, the modularity of CoK allows its application to various LLMs and different formats of knowledge sources, which addresses important challenges, including privacy concerns, knowledge source reliance, and rapid information updates.

## ACKNOWLEDGMENTS

This work was substantially supported by DAMO Academy through DAMO Academy Research Intern Program. This research/project is supported by the National Research Foundation, Singapore under its AI Singapore Programme (AISG Award No: AISG-PhD/2021-01-001). Soujanya Poria is supported by the grant AISG3-GV-2023-010.

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

## A    PROMPTS USED IN DIFFERENT METHODS

### A.1    CHAIN-OF-KNOWLEDGE (HOTPOTQA)

#### A.1.1    REASONING GENERATION

**Q:** This British racing driver came in third at the 2014 Bahrain GP2 Series round and was born in what year
**A:** First, at the 2014 Bahrain GP2 Series round, DAMS driver Jolyon Palmer came in third. Second, Jolyon Palmer (born 20 January 1991) is a British racing driver. The answer is 1991.

**Q:** [Question]
**A:**

#### A.1.2    KNOWLEDGE DOMAIN SELECTION

Follow the below example, select relevant knowledge domains from Available Domains to the Q.
Available Domains: factual, medical, physical, biology
**Q:** This British racing driver came in third at the 2014 Bahrain GP2 Series round and was born in what year
**Relevant domains:** factual
**Q:** Which of the following drugs can be given in renal failure safely?
**Relevant domains:** medical
**Q:** Which object has the most thermal energy?
**Relevant domains:** factual, physical
**Q:** Is the following trait inherited or acquired? Barry has a scar on his left ankle.
**Relevant domains:** biology

**Q:** [Question]
**Relevant domains:**

#### A.1.3    RATIONALE CORRECTION

Strictly follow the format of the below examples. The given sentence may have factual errors, please correct them based on the given external knowledge.
**Sentence:** the Alpher-Bethe-Gamow paper was invented by Ralph Alpher.
**Knowledge:** discoverer or inventor of Alpher-Bethe-Famow paper is Ralph Alpher.
**Edited sentence:** the Alpher-Bethe-Gamow paper was invented by Ralph Alpher.

**Sentence:** Ralph Alpher was advised by Hans Bethe.
**Knowledge:** doctoral advisor of Ralph Alpher is George Gamow.
**Edited sentence:** Ralph Alpher was advised by George Gamow.

**Sentence:** [Ratioanle]
**Knowledge:** [Knowledge]
**Edited sentence:**

#### A.1.4    NEXT RATIONALE GENERATION

**Q:** This British racing driver came in third at the 2014 Bahrain GP2 Series round and was born in what year
**A:** First, at the 2014 Bahrain GP2 Series round, DAMS driver Jolyon Palmer came in third. Second, Jolyon Palmer (born 20 January 1991) is a British racing driver. The answer is 1991.

**Q:** [Question]
**A:** First, [Corrected first rationale]. Second,

### A.1.5 ANSWER CONSOLIDATION

**Q:** This British racing driver came in third at the 2014 Bahrain GP2 Series round and was born in what year
**A:** First, at the 2014 Bahrain GP2 Series round, DAMS driver Jolyon Palmer came in third. Second, Jolyon Palmer (born 20 January 1991) is a British racing driver. The answer is 1991.

**Q:** [Question]
**A:** First, [Corrected first rationale]. Second, [Corrected second rationale]. The answer is

### A.2 COT, COT-SC

**Q:** This British racing driver came in third at the 2014 Bahrain GP2 Series round and was born in what year
**A:** First, at the 2014 Bahrain GP2 Series round, DAMS driver Jolyon Palmer came in third. Second, Jolyon Palmer (born 20 January 1991) is a British racing driver. The answer is 1991.

**Q:** [Question]
**A:**

### A.3 VERIFY-AND-EDIT

### A.3.1 VERIFYING QUESTION GENERATION

Write a question that asks about the answer to the overall question.
**Overall Question:** The Sentinelese language is the language of people of one of which Islands in the Bay of Bengal?
**Answer:** The language of the people of North Sentinel Island is Sentinelese.
**Question:** What peopleś language is Sentinelese?

**Overall Question:** [Question]
**Answer:** [Rationale]
**Question:**

### A.3.2 VERIFYING ANSWER GENERATION (RATIONALE EDITING)

Barnes House (born 20 January 1969) is a British racing driver, currently driving for Renault Sport F1 Team in the Formula One World Championship.
Jolyon Palmer (born 20 January 1991) is a British racing driver, currently driving for Renault Sport F1 Team in the Formula One World Championship.
Ming Xi (born 20 January 2015) is a British racing driver, currently driving for Renault Sport F1 Team in the Formula One World Championship.
The 2014 Bahrain GP2 Series round was a pair of motor races held on 6 and 7 April 2014 at the Bahrain International Circuit in Sakhir, Bahrain as part of the GP2 Series. Julián Leal finished second for the Carlin team and DAMS driver Jolyon Palmer came in third.
**Q:** This British racing driver came in third at the 2014 Bahrain GP2 Series round and was born in what year
**A:** This British racing driver came in third at the 2014 Bahrain GP2 Series round and was born in 1991.

Knowledge
**Q:** [Verifying question]
**A:**

## B  FURTHER EXPERIMENT DETAILS

### B.1  FETAQA

Although CoK and several baseline methods including CoT-SC and VE rely on self-consistency for other tasks, we note that self-consistency is not applicable for FeTaQA as it is an open-ended generation task. As a result, it is possible to have near-equivalent generations that are nevertheless not exact matches, and self-consistency becomes less useful. Therefore, we do not use self-consistency for VE and CoK, instead opting to retrieve from external knowledge sources for every question in FeTaQA. We also do not include CoT-SC results for FeTaQA in Table 2.

## C  QUERY EXECUTION OF KNOWLEDGE SOURCES

### C.1  WIKIDATA (SPARQL)

As shown in Table 1, the SPARQL query generated by AQG contains entity and relation spans. To make the query executable, we conduct entity linking, replacing the spans with entity and relation IDs. We utilize the GENRE model for entity linking (Cao et al., 2021). GENRE is the first system that retrieves entities by generating their unique names in an autoregressive fashion. Consequently, GENRE is capable of performing entity linking without ambiguities. Next, the query is executed on Wikidata to retrieve the results. Finally, we transform the reasoning step and the results into a natural sentence format, which serves as the final supporting knowledge.

### C.2  WIKIPEDIA (NATURAL SENTENCE)

We directly query generated natural language sentence within the domain `wikipedia.org`.

### C.3  TABLE (SQL)

Given a generated SQL query, we execute the query on the given table to obtain the result, which may be a single value or a sub-selection of the table. Thereafter, we consolidate the query result with the original question which is provided to the LLM for generating the final answer. As the query may be inaccurate in some cases, we also provide the original table to the LLM when generating the final answer.

### C.4  FLASHCARD (NATURAL SENTENCE)

Given a medical reasoning step, AQG generates a sentence of relevant medical knowledge as the query. Subsequently, we compare the embeddings of this query with sentences from the Medical Flashcards knowledge base and select the sentence with the highest cosine similarity as the final supporting knowledge. Hence, this ensures that the supporting knowledge is factually correct.

### C.5  UPTODATE (NATURAL SENTENCE)

We directly query generated natural language sentence within the domain `uptodate.com`, which is an authoritative medical website.

### C.6  SCIENCEQA PHYSICS (NATURAL SENTENCE)

Given a physics reasoning step, AQG generates a sentence of relevant physics knowledge as the query. Subsequently, we compare the embeddings of this query with sentences from the ScienceQA Physics knowledge source and select the sentence with the highest cosine similarity as the final supporting knowledge. Hence, this ensures that the supporting knowledge is factually correct.

### C.7  PHYSICSCLASSROOM (NATURAL SENTENCE)

We directly query generated natural language sentence within the domain `physicsclassroom.com`, which is an authoritative physics website.

Table 8: Query language, AQG model, and training datasets of each knowledge source. Knowl. stands for knowledge. Lang. stands for language.

| Knowl. Domain | Knowl. Source | Query Lang. | AQG Model | Dataset Source | Train. Set | Eval. Set |
|---|---|---|---|---|---|---|
| Factual | Wikidata | SPARQL | LLaMA-2-7B-LoRA | LC-quad & KQA-pro | 19,010 | 4,779 |
| Factual | Wikipedia | n.s. | gpt-3.5-turbo-0613 | - | - | - |
| Factual | Table | SQL | gpt-3.5-turbo-0613 | - | - | - |
| Medical | Flashcard | n.s. | LLaMA-2-7B-LoRA | Medical Flashcard | 34,000 | - |
| Medical | UpToDate | n.s. | gpt-3.5-turbo-0613 | - | - | - |
| Physics | ScienceQA Physics | n.s. | LLaMA-2-7B-LoRA | ScienceQA Physics | 810 | - |
| Physics | Physicsclassroom | n.s. | gpt-3.5-turbo-0613 | - | - | - |
| Biology | ScienceQA Biology | n.s. | LLaMA-2-7B-LoRA | ScienceQA Physics | 1,596 | - |
| Biology | CK-12 | n.s. | gpt-3.5-turbo-0613 | - | - | - |

## C.8   SCIENCEQA BIOLOGY (NATURAL SENTENCE)

Given a biology reasoning step, AQG generates a sentence of relevant biology knowledge as the query. Subsequently, we compare the embeddings of this query with sentences from the ScienceQA Biology knowledge source and select the sentence with the highest cosine similarity as the final supporting knowledge. Hence, this ensures that the supporting knowledge is factually correct.

## C.9   CK-12 (NATURAL SENTENCE)

We directly query generated natural language sentence within the domain `ck12.org/c/biology/`, which is an authoritative biology website.

## D   ADAPTIVE QUERY GENERATOR

### D.1   WIKIDATA (SPARQL)

#### D.1.1   INSTRUCTION-TUNING DATASET

To create the instruction-tuning dataset, we utilize a filtered version of LC-quad (Trivedi et al., 2017) and KQA-pro (Cao et al., 2022) datasets. This dataset consists of natural questions as inputs and their corresponding SPARQL queries as outputs. Before training, we replace the entity and relation IDs in the SPARQL queries with entity and relation spans. This modification allows the model to learn the semantic meaning of the SPARQL queries more effectively. During the inference phase, we utilize entity linking to convert the spans back into IDs. The size of the dataset is provided in Table 8.

#### D.1.2   DATA EXAMPLE

Below is an instruction that describes a task, paired with an input that provides further context. Write a response that appropriately completes the request.
**### Instruction:**
Generate a correct SPARQL query that returns the answer of the following question. Generate four incorrect SPARQL queries of different types.
**### Input:**
What periodical literature does Delta Air Lines use as a mouthpiece?
**### Output:**
Correct query: select distinct ?obj where { wd:/Delta Air Lines/ wdt:/house publication/ ?obj . ?obj wdt:/instance of/ wd:/periodical/ }
Incorrect query 1: select distinct ?obj where { Delta Air Lines wdt:/house publication/ ?obj . ?obj wdt:/instance of/ wd:/periodical/ }
Incorrect query 2: select distinct ?obj { wd:/Delta Air Lines/ wdt:/house publication/ ?obj . ?obj wdt:/instance of/ wd:/magazine/ }
Incorrect query 3: select distinct ?obj where { wd:/Delta Airlines/ wdt:/house publication/ ?obj . ?obj wdt:/instance of/ wd:/periodical/ }

Incorrect query 4: select distinct ?obj where { ?obj wdt:/instance of/ wd:/magazine/ . wd:/Delta Air Lines/ wdt:/house publication/ ?obj . }

## D.2 FLASHCARDS (NATURAL SENTENCE)

### D.2.1 INSTRUCTION-TUNING DATASET

We employ a natural sentence format for querying Medical knowledge. To instruction-tune our AQG specifically for this purpose, we utilize the Medical Flashcards dataset (Han et al., 2023). This dataset consists of question-answering pairs covering various subjects in the medical source, such as anatomy, physiology, pathology, and pharmacology. It contains summaries and mnemonics of crucial medical concepts, making it an ideal choice for instruction-tuning the AQG to effectively handle medical knowledge queries. The size of the dataset is provided in Table 8.

### D.2.2 DATA EXAMPLE

Answer the question truthfully.
**### Instruction:**
Answer this question truthfully.
**### Input:**
What is the relationship between very low Mg2+ levels, PTH levels, and Ca2+ levels?
**### Output:**
Very low Mg2+ levels correspond to low PTH levels which in turn results in low Ca2+ levels.

## D.3 SCIENCEQA PHYSICS (NATURAL SENTENCE)

### D.3.1 INSTRUCTION-TUNING DATASET

To instruction-tune our AQG for physics knowledge, we utilize the physics segment of the ScienceQA dataset (Lu et al., 2022). Each entry in this dataset consists of a question, options, context, answer, lecture, and explanation. The lecture contains necessary knowledge to answer the question. We use the question and the options as input and the lecture as the output for instruction-tuning the model.

### D.3.2 DATA EXAMPLE

Answer the question truthfully.
**### Instruction:**
Answer this question truthfully.
**### Input:**
The objects are identical except for their temperatures. Which object has less thermal energy? Choose from: a 300-gram glass of water at a temperature of 75°F, a 300-gram glass of water at a temperature of 80°F.
**### Output:**
The two glasses of water have the same mass but different temperatures. Since the 75°F glass of water is colder than the 80°F glass of water, it has less thermal energy.

## D.4 SCIENCEQA BIOLOGY (NATURAL SENTENCE)

### D.4.1 INSTRUCTION-TUNING DATASET

To instruction-tune our AQG for biology knowledge, we utilize the biology segment of the ScienceQA dataset (Lu et al., 2022).

### D.4.2 DATA EXAMPLE

Answer the question truthfully.
**### Instruction:**
Answer this question truthfully.

Table 9: Examples of generated query, execution results, and formatted knowledge for rationales of each query language. Knowl. stands for knowledge.

| SPARQL | |
|---|---|
| Rationale | Souleyman Sané's son, Leroy Sané, is a professional football player. |
| Generated query | SELECT ?answer WHERE { wd:/Souleymane Sané/ wdt:/child/ ?answer . } |
| Execution results | Leroy Sané |
| Formatted knowl. | The fact entity of the sentence "Souleyman Sané's son, Leroy Sané, is a professional football player" is Leroy Sané. |

| SQL | |
|---|---|
| Rationale | Does Oklahoma have any indoor football teams? |
| Generated query | SELECT * FROM table WHERE Type = 'Indoor Football'; |
| Execution results | |

| Club | Type | Venue |
|---|---|---|
| Oklahoma Defenders | Indoor Football | Tulsa Convention Center |

| | |
|---|---|
| Formatted knowl. | [['Club', 'Type', 'Venue'], ['Oklahoma Defenders', 'Indoor Football', 'Tulsa Convention Center']] |

| Natural Sentence | |
|---|---|
| Rationale | Splenomegaly is a condition in which the spleen is enlarged. |
| Generated query | What conditions may feature splenomegaly? |
| Execution results | Normocytic anemia with extravascular hemolysis is associated with enlargement of the spleen. |
| Formatted knowl. | Normocytic anemia with extravascular hemolysis is associated with enlargement of the spleen (splenomegaly), as the spleen plays a role in removing damaged red blood cells from circulation. |

### Input:
In a group of sheep, some individuals have white wool and others have black wool. In this group, the gene for the wool color trait has two alleles. The allele L is for white wool, and the allele l is for black wool. Flicka, a sheep from this group, has white wool. Flicka has one allele for white wool and one allele for black wool. Based on this information, what is Flicka's phenotype for the wool color trait? Choose from: Ll, white wool.
### Output:
An organism's phenotype for a trait is its observable version of that trait. Flicka's observable version of the wool color trait is white wool. So, Flicka's phenotype for the wool color trait is white wool.

## D.5 EXAMPLES OF EACH QUERYING LANGUAGE

We include examples of generated query, execution results, and formatted knowledge for rationales of each query language in Table 9.

## D.6 CONTRASTIVE INSTRUCTION-TUNING

We implement a simple approach to train the model for SPARQL with a contrastive objective, where the correct query and wrong queries are modeled autoregressively in the same sequence. Concretely, given a sequence $x$ which includes the input tokens, correct query tokens and wrong query tokens, the query model is trained with the log-likelihood loss:

$$\log p(x) = \log \prod_{i=1}^{n} \mathbf{1}(x_i) p(x_i | x_{<i}) \qquad (1)$$

where $\mathbf{1}(x_i) = 1$ if the i-th token $x_i$ is part of a query and 0 otherwise.

## D.7 TRAINING DETAILS

We employ Llama-2 (`meta-llama/Llama-2-7b-hf`) as the base model. We utilize LoRA for parameter-efficient fine-tuning, and load the weights in `8-bit` format. For each knowledge source, the model is trained for 3 epochs, utilizing an `NVIDIA A40` GPU. We maintain a training batch size of 32, with a gradient accumulation step set at 2. All the other parameters are left at their default values.

Table 10: Details of the evaluation datasets.

| Domain | Dataset | # of Samples |
|---|---|---|
| Factual | FEVER | 1000 |
| Factual | HotpotQA | 308 |
| Factual | FeTaQA | 500 |
| Medical | MedMCQA | 146 |
| Physics | MMLU Physics | 253 |
| Biology | MMLU Biology | 454 |

Table 11: Evaluation of domain selection performance.

| Domain | Precision | Recall | F1 |
|---|---|---|---|
| Factual | 96.0% | 96.0% | 96.0% |
| Medical | 94.3% | 96.1% | 95.2% |
| Physics | 89.9% | 100.0% | 94.6% |
| Biology | 100.0% | 92.8% | 96.2% |

## E   EVALUATION DATASETS

The evaluation datasets collect datasets from four different domains, including factaul, medical, physics, and biology. Details of the dataset are in Table 10. We adopt exact match as the evaluation metrics for HotpotQA, which is a more strict evaluation.

## F   ANALYSIS

### F.1   DOMAIN SELECTION

As CoK relies on selecting relevant knowledge domains, it is important that the domain selection step is of high quality. Hence, we randomly sample 50 questions from each domain and compare the predicted domains with our manually annotated domains. As each question may be relevant for more than one domain, we report the precision, recall, and F1 scores. As shown in Table 11, we find that while the domain selection is not perfect, the overall F1 scores are more than 94% across all the domains. Hence, we believe that the current domain selection process is adequate.

### F.2   MODELS OF ADAPTIVE QUERY GENERATOR

Table 12 demonstrates the performances of ChatGPT and instruction-tuned LlaMA-2-7B on SQL and SPARQL generation. SPARQL is evaluated on 4,779 samples from LC-quad and KQA-pro. SQL is evaluated on 15,900 samples from WikiSQL and we use the exact-match metric to evaluate the generated queries with gold queries.

## G   DISCUSSION OF LIMITATIONS

**Knowledge Sources**   As CoK relies on external knowledge sources, there are some ethical implications. Notably, LLMs using CoK may still generate inaccurate information if the knowledge sources contain unreliable information. Hence, this could cause misinformation or manipulation of public opinion. Another limitation is that there may be conflict between different knowledge sources in theory. To address the two limitations, we selected authoritative knowledge sources such as Wikidata which are unlikely to contain inaccurate or conflicting information. As a result, the risk from the knowledge sources are reduced.

**Knowledge Retrieval**   On the other hand, CoK may not produce useful outputs if the knowledge retrieval step is unable to retrieve facts that are relevant to the given question. However, we believe

Table 12: Performances of ChatGPT and instruction-tuned LlaMA-2-7B on SQL and SPARQL generation.

| Method | SQL Eval. Acc. | SPARQL Eval. Acc. |
|---|---|---|
| ChatGPT | 57.1% | 8.9% |
| Finetuned Model | 38.6% | 41.1% |

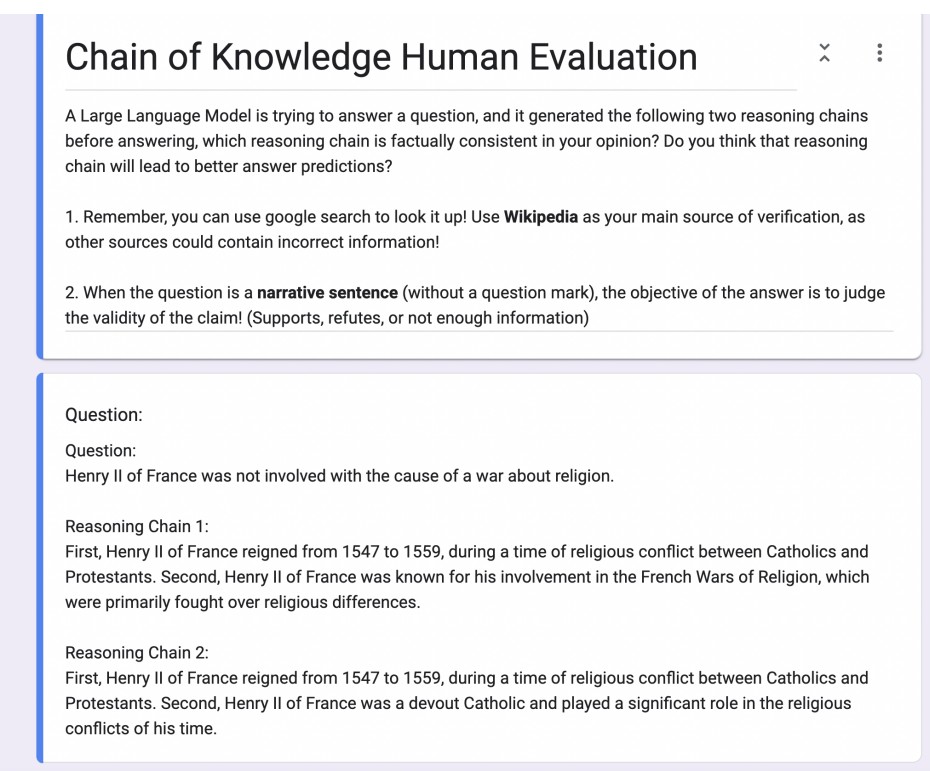

Figure 4: Human evaluation instructions.

that this is a general limitation of retrieval methods, as retrieval results inevitably contain some noise due to lack of relevant data or inaccurate queries. To address this challenge, we have designed the CoK framework to be modular and flexible. Hence, the adaptive query generator models can be easily swapped for other models that may be more suitable for the given task. Rather than focusing on using specific query generator models, our focus is that heterogeneous knowledge sources can be effectively incorporated with LLMs to improve their factual correctness and performance on knowledge-intensive tasks.

**Reasoning Capability of LLMs**    As CoK relies on the reasoning capability of LLMs, failure cases may stem from reasoning failures of LLMs. We believe this is a general limitation of generative models, as LLMs inevitably generate reasoning errors. Case studies of such failures can be found in Appendix H.2. To address this challenge, CoK is designed to be modular and flexible. And the black-box LLM can be easily swapped for more advanced models possessing enhanced reasoning capabilities.

Figure 5: Human evaluation questions.

## H HUMAN STUDY

### H.1 SETUP

In the human study, volunteers are proficient English speakers in relevant disciplines. The instruction specifically asks the volunteers to verify their knowledge from Google, especially the Wikipedia data source. The specific instructions given to the users are shown in 4. The questions given to the users are shown in 5. First, the user is asked which reasoning chain is factually consistent in his/her opinion. Here, we use a direct assessment rather than a comparative measure (for example, is one more factually correct than the other). Intuitively, factual consistency should not be "more" or "less". Similar direct measures are also preferred by the community, such as the direct assessment in Machine Translation (Kocmi et al., 2022; Graham et al., 2017). If they are both incorrect or both correct, the user could choose "Tie". Then, the user is asked whether he/she thinks the better reasoning chain will lead to better answer predictions. In scenarios where the user answers "Tie" to the first question, he/she will also answer "Tie" for the second question.

For evaluation, 100 samples are randomly chosen from HotpotQA and Fever datasets. The order of the reasoning chains (produced by CoK or CoT-SC) is randomly perturbed for each question.

### H.2 EXAMPLES

As mentioned in section 5.3, even when we improve the factual consistency of the CoTs, the outputs could still be false due to LLM's reasoning errors. We copy three such examples below:

**Example 1:**

**Prompt:** [3-shot CoT prompt]

Q: Anne Sullivan was born in June of 1866.

A: First, Anne Sullivan was born on April 14, 1866 in Feeding Hills, Agawam, Massachusetts, United States. Second, Anne Sullivan was born on April 14, 1866 in Feeding Hills, Agawam, Massachusetts, United States. The answer is

**ChatGPT:** SUPPORTS.

**Example 2:**

**Prompt:** [3-shot CoT prompt]

Table 13: Details on costs (by tokens).

| Method | Dataset | tokens |
|---|---|---|
| ReAct | HotpotQA | 1638 |
| Verify-and-Edit | HotpotQA | 630 |
| CoK | HotpotQA | 787 |
| ReAct | FEVER | 848 |
| Verify-and-Edit | FEVER | 286 |
| CoK | FEVER | 329 |

Q: Practical Magic is based on a French novel that was written by Alice Hoffman.

A: First, Practical Magic is a 1998 American fantasy romantic drama film based on the 1995 novel of the same name by Alice Hoffman. Second, Alice Hoffman is an American author. The answer is

**ChatGPT:** SUPPORTS.

**Example 3:**

**Prompt:** [3-shot CoT prompt]

Q: Saturn Corporation has no other names.

A: First, The Saturn Corporation, also known as Saturn LLC, was a registered trademark established on January 7, 1985, as a subsidiary of General Motors. Second, There is no information available on any other names for Saturn Corporation, but it is also known as Saturn LLC. The answer is

**ChatGPT:** SUPPORTS.

In the first example, it is mentioned twice in the prompt that Anne Sullivan was born in April. However, the LLM still supports the claim that she was born in June. In the second example, the CoT specifies that the novel is American. However, ChatGPT overlooks the nationality and supports the claim that it is based on a French novel. In the third example, the CoT mentions repetitively that Saturn Corporation is also known as Saturn LLC. However, ChatGPT supports the claim that it has no other names.

These examples show that, even though the CoT is successfully improved in terms of factual consistency, the final answer may still be incorrect due to reasoning errors inherent to LLM itself. In the human study for wrong predictions, 44% of the time humans claim that CoK still generates improved CoTs. Among these 44% instances, 73% of the time humans think these CoTs should have led to better answers.

## I  COST ANALYSIS

As CoK always edits instances below a certain consistency threshold, there is a cost advantage compared to other methods such as ReAct. The costs are on par with methods such as Verify-and-Edit.

A table of the costs is shown in 13. The costs are calculated based on tokens used per instance. Overall, the costs for CoK are on par with Verify-and-Edit. The extra costs are incurred by the dynamic knowledge editing stage, which is shown to boost performance in the main results. CoK also costs much less than ReAct, incurring only around 40% of ReAct's costs. Specifically, it costs 787 compared to 1638 for HotpotQA, and 329 compared to 848 for FEVER.

The API cost for `gpt-3.5-turbo` is currently $0.0015 / 1K tokens for input, and $0.002 / 1K tokens for output.

For details of the cost calculations, as the output length is the same for all methods, we only calculate the input tokens. Following the original ReAct paper(Yao et al., 2023), we calculate based on 3-shot prompts for FEVER and 6-shot prompts for HotpotQA. Verify-and-Edit and CoK tokens per instance are calculated based on the CoT-SC threshold, which results in editing 86 out of 308 instances for

HotpotQA 6-shot, and 127 out of 1,000 instances for FEVER 3-shot. The plain ReAct method, on the other hand, applies the ReAct prompt to every instance.

