# OpenReview forum: "Chain-of-Knowledge: Grounding Large Language Models via Dynamic Knowledge Adapting over Heterogeneous Sources"
_ICLR.cc/2024/Conference — ICLR 2024 poster_

### Official Review · Reviewer_TzUb · 2023-10-28

**Soundness:** 3 good
**Presentation:** 3 good
**Contribution:** 3 good
**Rating:** 6
**Confidence:** 3

**Summary:**

This paper proposed chain-of-knowledge (CoK), a novel framework that augments LLMs by dynamically incorporating grounding information from heterogeneous sources. The CoK framework consists of three stages, reasoning preparation, dynamic knowledge adapting, and answer consolidation. This paper provided rich experiments and analysis.

**Strengths:**

S1. This paper proposed chain-of-knowledge (CoK), a novel framework to enhance the factual correctness of LLMs with heterogeneous knowledge sources. CoK can dynamically select one or multiple knowledge sources based on the types of questions.

S2. Cok participates in Cot which modifies incorrect answers in each reasoning step by introducing external knowledge, thereby generating the correct answer.

S3. This paper performs extensive experiments on knowledge-intensive tasks spanning a range of domains, including factual, medical, physical, and biological.

**Weaknesses:**

This paper bears a high resemblance to VE with the main distinction being the inclusion of multiple knowledge sources, which might be considered a relatively ordinary contribution. The AQG retrieves relevant knowledge from multiple knowledge sources by converting questions into SPARQL, SQL, and natural language queries. The SPARQL query generator is fine-tuned based on question-SPARQL pairs. I think this training process may make it challenging for AQG to generate SPARQL queries for complex and compositional questions. Because it cannot guarantee that the sub-questions derived from the CoT decomposition are all simple.

**Questions:**

Q1. The authors specifically evaluated the proposed framework in experiments, with a focus on the biology, medical, and physics domains, showing better results compared to the VE method. The knowledge sources used by the author include Wikidata, medical Flashcard, UpToDate, ScienceQA Physics, and ScienceQA Biology. I'm wondering if the authors also incorporated these knowledge sources into the VE method. If not, I believe such a comparison would be unfair.

Q2. LLMs generate multiple results based on self-consistency, if the consistency falls below a threshold, the proposed method is initiated to correct the answer. I'm curious about how often such a process needs to be introduced in practical experiments. Which of the three extraction methods (natural language, SPARQL, and SQL) do the authors consider to be more reliable for LLMs?

---

> ### Author Response · Authors · 2023-11-17
> **Response to Reviewer TzUb**
>
> Dear reviewer TzUb,
>
> Thank you for your recognition of our extensive experiments and analysis. We would like to address your queries and concerns here.
>
> >This paper bears a high resemblance to VE with the main distinction being the inclusion of multiple knowledge sources, which might be considered a relatively ordinary contribution.
>
> The key contributions of our work are the following:
> 1. Querying diverse heterogeneous knowledge sources with appropriate queries and aggregating answers from those in an effective manner posits unique scientific challenges and were not addressed in prior work. To the best of our knowledge, we are the first to enhance the factual correctness of LLMs with heterogeneous knowledge sources.
> 2. To the best of our knowledge, we are the first to progressively correct the rationales, ensuring that inaccuracies from preceding rationales do not propagate into the subsequent steps. Prior studies such as VE and ReAct leave errors from prior steps in the prompt, causing potential noise.
> 3. We propose AQG, which is versatile and can seamlessly transition between fine-tuned models and black-box LLMs, enabling scalability of knowledge sources.
> 4. Extensive experiments prove the effectiveness of CoK across ranges of domains, including factual, medical, physical, and biological.
>
> >I think this training process may make it challenging for AQG to generate SPARQL queries for complex and compositional questions. Because it cannot guarantee that the sub-questions derived from the CoT decomposition are all simple.
>
> We used an AQG trained with data whose logical granularity is on par with the CoT rationales. For example, for WikiData, both training data and CoT rationales only contain one entity and one relation in each sentence. This enables the AQG to formulate more precise queries.
>
> >I'm wondering if the authors also incorporated these knowledge sources into the VE method. If not, I believe such a comparison would be unfair.
>
> Yes. In our baseline results for VE, the retrieval system used is Google Search, which includes all the knowledge that CoK can access.
>
> >I'm curious about how often such a process needs to be introduced in practical experiments.
>
> As mentioned in Appendix I, 86 out of 308 instances for HotpotQA 6-shot, and 127 out of 1000 instances for FEVER 3-shot pass through the knowledge adapting stage.
>
> >Which of the three extraction methods (natural language, SPARQL, and SQL) do the authors consider to be more reliable for LLMs?
>
> Could you please clarify your question? If you are inquiring about which query format is most reliable for LLMs to produce, then the query format would depend on which knowledge source we are retrieving from. For example, for knowledge graphs such as wikidata, SPARQL is used. For natural sentence knowledge sources such as Medical Flashcard, natural language is used. Using different query languages that do not match the format of the knowledge source will likely result in suboptimal performances.

---

> > ### Comment · Reviewer_TzUb · 2023-11-21
> >
> > I have read the author's rebuttal. My concerns about this paper have been resolved. In short, this paper proposed a novel framework to enhance the factual correctness of LLMs with heterogeneous knowledge sources. I would like to keep my current scores.

---

### Official Review · Reviewer_SrPL · 2023-10-29

**Soundness:** 3 good
**Presentation:** 3 good
**Contribution:** 3 good
**Rating:** 6
**Confidence:** 3

**Summary:**

The paper introduces the chain-of-knowledge (CoK) framework designed to augment the performance of LLMs by reducing instances of information hallucination and improving factual correctness. The CoK operates in three stages: reasoning preparation, dynamic knowledge adaptation, and answer consolidation. A distinctive feature of CoK is its ability to dynamically tap into diverse knowledge sources, including structured databases like Wikidata, using an adaptive query generator (AQG) capable of generating varied query languages, such as SPARQL, SQL, and natural language queries. The framework emphasizes progressive correction of the generated rationales, minimizing the risk of error propagation across reasoning steps. Experimental evaluations indicate that CoK enhances the performance of LLMs on knowledge-intensive tasks across different domains, surpassing the chain-of-thought (CoT) baseline by an average of 4.3%.

**Strengths:**

- CoK addresses the inherent limitations of existing methods by leveraging diverse knowledge sources, both structured and unstructured, improving the accuracy and factual correctness of LLM outputs.
- The adaptive query generator (AQG) showcases versatility, enabling seamless transition between specialized models and generic LLMs, allowing effective querying across different knowledge source formats.
- Progressive rationale correction reduces the risk of error propagation, ensuring more reliable generation of answers.
- Comprehensive experiments covering various knowledge domains, demonstrating a consistent performance boost over the CoT baseline.

**Weaknesses:**

- While the paper emphasizes the use of diverse knowledge sources, it might benefit from a more explicit discussion on the scalability and efficiency of the framework as the volume and variety of sources grow.
- The framework's dependence on the AQG's capability to generate effective queries for all types of knowledge sources might be a potential bottleneck, especially for highly specialized domains.
- Although the paper mentions improvement over the CoT baseline, deeper insights into the limitations and areas where CoK might not perform optimally would be beneficial. Except for the limitations of “Knowledge Sources” and “Knowledge Retrieval” discussed in Appendix G.

**Questions:**

- How does the CoK framework handle situations where knowledge sources provide conflicting information? How are you going to solve it?
- What are the computational overheads introduced by the AQG and the dynamic knowledge adaptation process, especially when querying multiple heterogeneous sources?
- Have there been considerations or plans to extend the CoK framework to accommodate real-time or streaming knowledge sources?
- Can you provide insights into the training and fine-tuning process of the AQG, especially its adaptability across different query languages and knowledge sources?

---

> ### Author Response · Authors · 2023-11-17
> **Response to Reviewer SrPL (1/2)**
>
> Dear reviewer SrPL,
>
> Thank you for your recognition of our comprehensive experiments and contributions! We would like to address your queries and concerns here.
>
> >While the paper emphasizes the use of diverse knowledge sources, it might benefit from a more explicit discussion on the scalability and efficiency of the framework as the volume and variety of sources grow.
> >What are the computational overheads introduced by the AQG and the dynamic knowledge adaptation process, especially when querying multiple heterogeneous sources?
>
> As sources grow, CoK will remain computationally efficient due to the following reasons:
> 1. The knowledge domain selection step enhances the efficiency of knowledge retrieval by ensuring only the most pertinent knowledge sources are selected for retrieval.
> 2. As highlighted in Section 1, AQG is versatile and can either be a fine-tuned model or an off-the-shelf LLM. The fine-tuning process is a one-time effort. And using off-the-shelf LLMs only requires prompt engineering. This simplicity facilitates the scaling of both the volume and variety of knowledge sources.
>
> The computational overheads of CoK are on-par with or less than existing retrieval augmentation methods:
> The inference of CoK is on par with Verify-and-Edit (VE) in terms of time and cost. Firstly, the inference time of CoK is similar to VE, with the only additional time required being for query generation. VE utilizes ChatGPT for this purpose. AQG can use either a fine-tuned LLaMA-2 or ChatGPT. The average inference time per instance for LLaMA-2 is 4 seconds, while for ChatGPT is 2 seconds. During inference, data retrieval from various knowledge sources is executed concurrently. As such, the maximum inference time overhead of CoK compared with VE per instance is only 2 seconds. Secondly, as shown in Table 12, the cost per instance of CoK is on par with VE as well. The extra costs are incurred by the dynamic knowledge editing stage, which is shown to boost performance in the main results.
>
> CoK also costs much less than ReAct, incurring only around 40% of ReAct’s costs. A detailed cost analysis can be found in Appendix I and Table 12.
>
> >The framework's dependence on the AQG's capability to generate effective queries for all types of knowledge sources might be a potential bottleneck, especially for highly specialized domains.
>
> As mentioned in the CoK framework (Section 3), AQG is a versatile component and can always be replaced by the SOTA model depending on the querying language. Furthermore, AQG can be either a fine-tuned model such as LLaMA-2 or a black-box LLM. For instance, as indicated in Table 11, when generating commonly-used querying languages like SQL, ChatGPT produces better results. However, for specialized domains that require specific querying languages, such as SPARQL, we find that a fine-tuned LLaMA-2 proves to be a more performant option.
>
> >Although the paper mentions improvement over the CoT baseline, deeper insights into the limitations and areas where CoK might not perform optimally would be beneficial. Except for the limitations of “Knowledge Sources” and “Knowledge Retrieval” discussed in Appendix G.
>
> Thank you for the suggestion! One potential limitation is that CoK relies on the reasoning capability of LLMs. Thus, failure cases may stem from reasoning failures of LLMs. Please refer to the Appendix G in our revised submission for more information. We also include three examples of these failure cases in Appendix H.2.
>
> >How does the CoK framework handle situations where knowledge sources provide conflicting information? How are you going to solve it?
>
> As highlighted in Appendix G, we only select highly authoritative knowledge sources which are unlikely to contain inaccurate or conflicting information. Additionally, we manually examined a random selection of 100 samples from HotpotQA and found no instances of conflicting information.
>
> >Have there been considerations or plans to extend the CoK framework to accommodate real-time or streaming knowledge sources?
>
> CoK supports real-time and streaming knowledge sources. For knowledge sources such as Wikidata, Wikipedia, UpToDate, PhysicsClassroom, and CK-12, knowledge is retrieved from their corresponding online websites. These websites are consistently maintained and updated in real-time. Furthermore, CoK supports static data, such as local tables, Flashcard, ScienceQA Physics, and ScienceQA Biology. We believe that this further demonstrates the versatility of CoK.

---

> > ### Author Response · Authors · 2023-11-17
> > **Response to Reviewer SrPL (2/2)**
> >
> > >Can you provide insights into the training and fine-tuning process of the AQG, especially its adaptability across different query languages and knowledge sources?
> >
> > Thank you for the suggestion!
> > For each knowledge source that requires specific training, we apply contrastive instruction-tuning to Llama-2 with LoRA as AQG. Details about constructing the dataset, including examples for each knowledge source, are provided in Appendix D.
> >
> > We have added more details about the AQG in Section 3 in the revised submission. This includes task-specific details such as providing the table schema for SQL, and how the domain of the training data corresponds to the respective knowledge source. Additionally, we have added more training details in Appendix D.6, including the base model, training epochs, batch size and other detailed settings. To ensure reproducibility, we will release our training code, data, and model weights.
> >
> > We also would like to highlight that certain AQG models are more suitable for specific querying languages as shown in Table 11. For example, off-the-shelf ChatGPT is more suitable for SQL query generation.  And our framework is flexible to support different fine-tuned models or off-the-shelf models as AQG.

---

> ### Comment · Reviewer_SrPL · 2023-11-22
> **Response to Authors and Chairs after Reviewing the Rebuttal**
>
> Dear Authors and Chairs,
>
> I have reviewed the rebuttal from authors, and really appreciate the authors' effort and further explanation on this work.
>
> Their response have addressed my concerns, and I recognize the versatility of robustness and generalizability of CoK. The main reason that I choose to keep my score is that, the additional contribution of this work compared with Verify-and-Edit (VE) [1] is anyhow limited.
>
> If this work can be further improved, for example, to propose a more general framework of refining reasoning process during training and after generation with adaptive knowledge retrieval, and provide more theoretical analysis, I think it will make this work much more solid and can better meet the ICLR acceptance criteria.
>
> [1] Verify-and-Edit: A Knowledge-Enhanced Chain-of-Thought Framework (ACL 2023)
>
> Best Wishes,
>
> Reviewer SrPL

---

> > ### Author Response · Authors · 2023-11-22
> > **Contributions of CoK compared to existing RAG methods**
> >
> > Dear Reviewer SrPL,
> >
> > Thank you for your recognition of the versatility of robustness and generalizability of CoK. We would like to address your concerns on the contributions of CoK compared to previous retrieval-augmented generation (RAG) methods, such as VE.
> >
> > 1. To the best of our knowledge, we are the first to **progressively correct the rationales**. As shown in Figure 1, error propagation could occur in existing RAG methods including VE and ReAct as they leave errors from prior steps in the prompt, causing potential noise. With progressive correction, CoK can effectively alleviate this issue.
> > 2. Querying diverse **heterogeneous knowledge sources** with appropriate queries and aggregating answers from those in an effective manner posits unique scientific challenges. To the best of our knowledge, we are the first to enhance the factual correctness of LLMs with heterogeneous knowledge sources. Previous RAG methods such as VE and ReAct can only retrieve unstructured knowledge (text-only), which limits the knowledge scope of the model.
> > 3. One of the major bottlenecks of existing RAG methods including VE is their retrieval capabilities. To address this issue, we  propose **AQG for effective knowledge retrieval**. AQG is designed to generate more accurate queries. AQG is versatile and can seamlessly transition between fine-tuned models and black-box LLMs, enabling scalability of knowledge sources.
> > 4. As shown in Table 1, VE could perform worse than baseline methods, especially in domain specific datasets. One identified problem in existing RAG methods is that the knowledge retrieved might not be related to the domain of the questions. CoK addresses this issue by **domain selection**, guaranteeing the knowledge is retrieved from the most pertinent sources. Extensive experiments prove the effectiveness of CoK across ranges of domains, including factual, medical, physical, and biological.
> >
> > We hope our clarification address your concerns on our contributions. If you have more questions or concerns, we would be more than happy to attend to them. Thank you.
> >
> > Best Regards,
> > Authors

---

### Official Review · Reviewer_bxPp · 2023-10-30

**Soundness:** 2 fair
**Presentation:** 2 fair
**Contribution:** 3 good
**Rating:** 6
**Confidence:** 4

**Summary:**

This paper proposes a chain-of-knowledge framework to augmenting LLM in question answering by dynamically retrieving the grounding information from multiple sources. The proposed method first gives the rationale in steps. Then the search query is generated by determining the domain involved in each rationale and the final query. Eventually, the rationale is corrected by the information obtained from the query to give a more plausible answer. The authors conducted experiments on multiple datasets and defeated the compared few-shot methods.

**Strengths:**

1. This article proposes a novel idea to reduce the hallucination of LLM by correcting each rationale in the chain of thought.
2. The proposed method considers structured and unstructured queries for heterogeneous data sources.
3. The authors conducted detailed experiments and analysis on multiple datasets from different domains.

**Weaknesses:**

1. The general outline of the suggested approach is somewhat understandable, but the crucial stages lack the necessary level of specificity. In my opinion, a key point in the effectiveness of the proposed method is how to construct accurate queries to retrieve grounding information, but the description in section 3 is still not clear enough. For example, how to build a high-quality question-sparql dataset to ensure the accuracy of this step. While the prompt used with the training loss function is provided in the appendix by the authors, I believe it is important for the main body of the paper to be self-contained, and critical steps like this should not solely rely on the information provided in the appendix.
2. How does the author determine the table or tables used when generating a SQL query for a rationale, and how can they ensure that the query generated by ChatGPT is valid and answerable if table schema information is not provided. Ditto for SPARQL.
3. Baseline performance on the hotpotQA dataset seems too poor. Existing work [1] has achieved an accuracy of 0.37 when reviewing ChatGPT.

[1] Zheng, Shen, Jie Huang, and Kevin Chen-Chuan Chang. "Why does chatgpt fall short in providing truthful answers." ArXiv preprint, abs/2304.10513 (2023).

**Questions:**

1. How does the author ensure that LLM can give faithful answers (query) when performing knowledge adaptation?
2. The proposed method leaves the decomposition of the question (i.e., the generation of each rationale) to ChatGPT. I am interested in whether this method is only applicable to chain-style question? For more complex questions, where the corresponding SPARQL contains more than one inference path, can the proposed method handle it?
3. When generating SQL (SPARQL) queries, how do the authors guarantee that the produced queries are valid?

---

> ### Author Response · Authors · 2023-11-17
> **Response to Reviewer bxPp (1/2)**
>
> Dear reviewer bxPp,
>
> Thank you for your time and suggestions. We would like to address your queries and concerns here.
>
> >In my opinion, a key point in the effectiveness of the proposed method is how to construct accurate queries to retrieve grounding information, but the description in section 3 is still not clear enough.
>
> Thank you for the suggestion! We have added more details about the adaptive query generator in Section 3. This includes task-specific details such as providing the table schema for SQL, and how the domain of the training data corresponds to the respective knowledge source.
>
> >How does the author determine the table or tables used when generating a SQL query for a rationale, and how can they ensure that the query generated by ChatGPT is valid and answerable if table schema information is not provided. Ditto for SPARQL.
>
> Following the existing formulation for the FeTaQA task, each question is provided with a specific table and the table schema. To ensure a successful query generation, the AQG should be aware of the structure and schema of the knowledge source. Therefore we always feed in the table schema.
>
> For SPARQL, no table schema is required. Instead, we use an AQG trained with data whose logical granularity is on par with the CoT rationales, i.e, both training data and CoT rationales only contain a single entity and relation in each sentence. This enables the AQG to formulate more precise queries.
>
> >Baseline performance on the hotpotQA dataset seems too poor. Existing work [1] has achieved an accuracy of 0.37 when reviewing ChatGPT.
>
> Following hotpotQA [2] and other previous work [3,4], we adopt the exact match metric for evaluating hotpotQA. However in [1], the evaluation for hotpotQA is partial match, a less stringent standard that typically yields higher scores. Below, we include results using partial match for hotpotQA for your consideration. In this case, CoK still outperforms over baseline methods.
>
> |       Method      | HotpotQA (E.M.) | HotpotQA (P.M.) |
> |:-----------------:|:---------------:|:---------------:|
> | Standard (3-shot) |      22.7%      |      32.5%      |
> |    CoT (3-shot)   |      29.9%      |      39.0%      |
> |  CoT-SC (3-shot)  |      30.8%      |      40.6%      |
> |    VE (3-shot)    |      31.8%      |      43.8%      |
> |  **CoK (3-shot)** |    **34.1%**    |    **47.1%**    |
> | Standard (6-shot) |      24.0%      |      33.4%      |
> |    CoT (6-shot)   |      34.4%      |      39.9%      |
> |  CoT-SC (6-shot)  |      33.4%      |      41.2%      |
> |    VE (6-shot)    |      34.4%      |      48.4%      |
> |  **CoK (6-shot)** |    **35.4%**    |    **49.7%**    |
>
>
>
> [1] Zheng, Shen, Jie Huang, and Kevin Chen-Chuan Chang. "Why does chatgpt fall short in providing truthful answers." ArXiv preprint, abs/2304.10513 (2023).\
> [2] Yang, Zhilin, et al. "HotpotQA: A dataset for diverse, explainable multi-hop question answering." In Proceedings of EMNLP, 2018.\
> [3] Zhao, Ruochen, et al. "Verify-and-edit: A knowledge-enhanced chain-of-thought framework." In Proceedings of ACL, 2023.\
> [4] Yao, Shunyu, et al. "React: Synergizing reasoning and acting in language models." In Proceedings of ICLR, 2023.

---

> ### Author Response · Authors · 2023-11-17
> **Response to Reviewer bxPp (2/2)**
>
> >How does the author ensure that LLM can give faithful answers (query) when performing knowledge adaptation?\
> >When generating SQL (SPARQL) queries, how do the authors guarantee that the produced queries are valid?
>
> We acknowledge that there is a possibility of AQG generating erroneous queries. However, we believe this is a common problem for all generative models. To mitigate this, we employ diverse techniques for query generation and  knowledge retrieval:
> 1. For query generation, AQG utilizes either black-box LLM or fine-tuned Llama-2. When using black-box LLM such as ChatGPT, we employ few-shot prompting and incorporate all necessary information in the prompt. For example, when generating SQL queries, the table schema along with data snippets are included in the prompt. When utilizing fine-tuned Llama-2, we ensure the logical granularity of AQG training data is on par with the decomposed CoT rationales. For example for WikiData, both training data and CoT rationales only contain one entity and one relation in one sentence. This enables the AQG to formulate more precise queries.
> 2. Our knowledge retrieval relies on highly authoritative knowledge sources. Therefore, even if the generated queries contain inaccuracies, the retrieved knowledge is faithful. As outlined in Appendix A.1.3, during knowledge editing, we also include a demonstration in the training prompt where the original rationale is unchanged. Thus, if no usable knowledge is retrieved, the baseline scenario would be where the original CoT rationale remains unchanged.
>
> As highlighted in Section 5.3, we provide both quantitative and human evaluations to demonstrate the effectiveness of our techniques. Results demonstrate that CoK enhances the factual accuracy of rationales, leading to more precise final answers.
>
>
> >I am interested in whether this method is only applicable to chain-style question? For more complex questions, where the corresponding SPARQL contains more than one inference path, can the proposed method handle it?
>
> The chain-of-knowledge framework utilizes chain-of-thought decomposition, which is not only beneficial for multi-hop QA (i.e., chain-style questions), but also questions requiring parallel reasoning (i.e., questions containing more than one inference path). Experiments on FEVER dataset support its effectiveness as FEVER is not multi-hop (chain-style), but contains claims that require composition of independent evidence from multiple sentences [5]. \
> Furthermore, we ensure that each rationale after CoT decomposition only contains a single entity and relation via prompt engineering as shown in Appendix A.1.1. As shown in Appendix D.1.2, the natural sentence input of the training data for AQG also contains a single entity and relation. This alignment in the complexity level of rationales and training data allows AQG to generate more accurate queries.
>
> [5] Thorne, James, et al. "FEVER: a large-scale dataset for fact extraction and VERification." In Proceedings of NAACL, 2018.

---

> > ### Author Response · Authors · 2023-11-22
> > **Thank you and further discussion**
> >
> > Dear reviewer bxPp,
> >
> > We would like to thank you for your insightful review. Following your valuable suggestions, we have updated our submission accordingly. Could we kindly enquire if our responses and adjustments have adequately resolved your concerns? We are more than happy to answer any further queries or concerns you may have. Thank you once again.
> >
> > Best Regards,
> >
> > Authors

---

> ### Comment · Reviewer_bxPp · 2023-11-22
>
> Thanks to the author for the reply. Some of my concerns are addressed. I expect that the authors will be able to explain the issues mentioned in more detail in the main body of the paper, including the validation of the query, and the choice of evaluation metric. I have updated the rating.

---

### Official Review · Reviewer_84zL · 2023-11-01

**Soundness:** 3 good
**Presentation:** 3 good
**Contribution:** 3 good
**Rating:** 6
**Confidence:** 4

**Summary:**

This is a very interesting paper on Chain of Knowledge and integrating the question answering from LLM with traditional source of knowledge such as KG to improve the overall accuracy of question answering in domain-specific text.
Main contributions:
1. Chain of Knowledge for factual correctness
2. Adaptive query generator for identifying knowledge source and converting queries in respective form
3. Progressive correction of rationale using CoK
4. Following domains have been chose for the work factual, medical, physical, and biological.

**Strengths:**

The paper is very well written.
The authors have considered good number of scenarios for knowledge, and carefully selected wikidata as the knowledge source, provided the limitation on retrieval of knowledge in general.
I enjoyed reading this paper.

**Weaknesses:**

Please see the questions section for more.

**Questions:**

* For the given SPARQL query SELECT ?answer WHERE { wd:/Souleymane Sane/ wdt:/child/ ?answer . ´ }, do you use some library to convert the entities/relation in their specific identifier form (entity linking), how do you know which is the correct identifier, in specific cases where similar natural language entities have two identifiers.
* Did you try domain specific questions with ChatGPT? Upon trying the example given in Figure 1 on ChatGPT, I could see the result as George Gamow. Maybe you should come up with an example where ChatGPT would hallucinate. Or if the idea here is to show open source LLM, then this example makes complete sense. Although it should be mentioned in this case which model is used (Llama2-7b-chat? or instruction tuned by yourself)
* It is possible that the knowledge injected through Wikidata doesn't answer the question asked, what would be done to answer the question in that case?
* Can you please explain how do you find specific relation for querying with an entity? There is a possibility of multiple entity attached with one relation. It is also possible that multiple relations connected with multiple entities, and it may be difficult to identify the relation depending on the question. What would be done in that case?

---

> ### Author Response · Authors · 2023-11-17
> **Response to Reviewer 84zL**
>
> Dear reviewer 84zL,
>
> Thank you for your recognition of our contributions! We would like to address your queries and concerns here.
>
> >do you use some library to convert the entities/relation in their specific identifier form (entity linking), how do you know which is the correct identifier, in specific cases where similar natural language entities have two identifiers.
>
> We employ the commonly used GENRE model [1] for entity linking, which effectively addresses the cases of ambiguous entities by generating unique entity identifiers. We have also included the details in Appendix C.1 in our revised submission.
>
> >Maybe you should come up with an example where ChatGPT would hallucinate.
>
> Thank you for your suggestion! In general, we observe that ChatGPT is more likely to hallucinate for domain-specific questions. Our chosen example highlights that even for questions needing common factual knowledge, ChatGPT can generate erroneous responses. We utilized gpt-3.5-turbo-0613  for our experiments, the latest version available at the time of experiments.
>
> In the revised submission, we have updated the example where the model failed to provide a correct answer. The CoT result in the paper can be reproduced using gpt-3.5-turbo-0613 with a temperature of 0, following the below prompt. Furthermore, even the latest gpt-3.5-turbo model generates erroneous responses.
>
> """\
> Strictly follow the format of the below examples, provide two rationales before answering the question.\
> Q: This British racing driver came in third at the 2014 Bahrain GP2 Series round and was born in what year?\
> A: First, at the 2014 Bahrain GP2 Series round, DAMS driver Jolyon Palmer came in third. Second, Jolyon Palmer (born 20 January 1991) is a British racing driver. The answer is 1991.
>
> Q: What band did Antony King work with that formed in 1985 in Manchester?\
> A: First, Antony King worked as house engineer for Simply Red. Second, Simply Red formed in 1985 in Manchester. The answer is Simply Red.
>
> Q: How many inhabitants were in the city close to where Alberta Ferretti’s studios was located?\
> A: First, Alberta Ferretti’s studio is near Rimini. Second, Rimini is a city of 146,606 inhabitants. The answer is 146,606.
>
> Q: What year was the Argentine actor who directed El Tio Disparate born?\
> A: First,\
> """
>
>
> >It is possible that the knowledge injected through Wikidata doesn't answer the question asked, what would be done to answer the question in that case?
>
> As illustrated in Appendix A.1.3, we include a demonstration in the training prompt where the original rationale is unchanged. Thus, if no usable knowledge is retrieved, the baseline scenario would be where the original CoT rationale remains unchanged (a fall-back strategy).
>
> We acknowledge that retrieval of irrelevant information (i.e., doesn’t answer the question asked) is a possibility. To mitigate this, we employed two strategies in this paper throughout the framework, leading to SOTA results compared to existing methods, such as ReAct and VE, as shown in Table 2.
> 1. We used an AQG trained with data whose logical granularity (number of entities and relations involved)  is on par with the CoT rationales. For example for WikiData, both training data and CoT rationales only contain one entity and one relation in one sentence. This enables the AQG to formulate more precise queries.
> 2. We incorporated multiple knowledge sources for each domain to get more validating evidence. The results in Table 4 confirm that using various sources enhances knowledge coverage.
>
>
>
> >Can you please explain how do you find specific relation for querying with an entity? There is a possibility of multiple entity attached with one relation. It is also possible that multiple relations connected with multiple entities, and it may be difficult to identify the relation depending on the question. What would be done in that case?
>
> As illustrated in Appendix A.1.1, every rationale after the reasoning preparation stage should comprise a single entity and relation. Similarly, Appendix D.1.2 indicates that each training data sample for SPARQL contains only one entity and one relation. This consistency ensures that the training data aligns with the rationale. Consequently, AQG can effectively generate more precise and disambiguous queries.
>
>
> [1] Nicola De Cao, Gautier Izacard, Sebastian Riedel, and Fabio Petroni. Autoregressive entity retrieval. In Proceedings of ICLR, 2021.

---

> > ### Author Response · Authors · 2023-11-22
> > **Thank you and further discussion**
> >
> > Dear reviewer 84zL,
> >
> > We would like to thank you for your insightful review. Following your valuable suggestions, we have updated our submission accordingly. Could we kindly enquire if our responses and adjustments have adequately resolved your concerns? We are more than happy to answer any further queries or concerns you may have. Thank you once again.
> >
> > Best Regards,
> >
> > Authors

---

> > > ### Author Response · Authors · 2023-11-23
> > > **Thank you and followup discussion**
> > >
> > > Dear reviewer 84zL:
> > >
> > > Thank you so much again for your valuable suggestions on:
> > >
> > > 1. Details on the model for entity linking.
> > > 2. Alternative example in Figure 1.
> > >
> > > We have carefully responded to your feedback and made adjustments in the submission to address your concerns. We would like to discuss this further with you to see if there are any unresolved questions. Thank you once again.
> > >
> > > Best Regards,
> > >
> > > Authors

---

### Author Response · Authors · 2023-11-17
**General response to all reviewers**

Dear reviewers,

We appreciate the valuable comments of all reviewers. We have uploaded a revised version of our paper that incorporates the suggestions in the reviews. And the changes are marked in **blue**. We believe the new version is clearer based on the valuable suggestions. Please let us know if our responses address your questions or concerns. We will be very happy to respond to further questions. Thank you all once again.

---

### Author Response · Authors · 2023-11-20
**Thanks to reviewers and further discussions**

Dear reviewers,

We would like to thank you for your valuable reviews and suggestions once again.
We have responded to your queries and concerns.
Following your insightful suggestions, we have revised our submission as well.
We are eager to engage in further discussions about these updates and are more than willing to address any additional questions you may have.
We sincerely appreciate your time and consideration.

Best Regards,

Authors

---

### Author Response · Authors · 2023-11-23
**Thanks to all reviewers and general summary**

Dear reviewers and chairs,

We would like to thank all the reviewers for their insightful and constructive reviews once again. We are grateful that reviewers found our paper is well written (**84zL**), novel (**bxPp**, **TzUb**), and conducts comprehensive experiments (**84zL**, **bxPp**, **SrPL**, **TzUb**). And we are delighted that subsequent discussions have successfully addressed your concerns (**bxPp**, **SrPL**, **TzUb**), and reviewer **bxPp** raised the score.

We would like to highlight our key contributions and summarize the enhancements we have made to our submission following the insightful suggestions of the reviewers.

The contributions of CoK compared to previous retrieval-augmented generation (RAG) methods, such as VE and ReAct can be summarized as follows:
1. To the best of our knowledge, CoK is the first to apply **progressive correction to rationales**, as depicted in Figure 1, mitigating error propagation seen in existing RAG methods.
2. CoK uniquely addresses the scientific challenges associated with querying and integrating diverse **heterogeneous knowledge sources** to improve the factual accuracy of LLMs. Previous RAG methods can only retrieve unstructured knowledge (text-only), which limits the knowledge scope of the model.
3. We introduce the **adaptive query generator (AQG) for enhanced retrieval efficiency**, which stands out in its improved query accuracy, flexibility and scalability when interacting with a variety of knowledge sources.
4. As shown in Table 1, we demonstrate how CoK, through **domain selection**, ensures relevance and improves performance consistently across various domains and datasets. Existing RAG methods such as VE could perform worse than baseline methods, especially in domain specific datasets.

Following the insightful suggestions of the reviewers, we have made the following enhancements to our submission:
1. (**84zL**) We include the details of the GENRE model for entity linking in Appendix C.1.
2. (**84zL**) We updated the example in Figure 1 with an example where gpt-3.5-turbo-0613 failed to provide a correct answer.
3. (**bxPp**) We have added more details about the adaptive query generator in Section 3. This includes task-specific details such as providing the table schema for SQL, and how the domain of the training data corresponds to the respective knowledge source.
4. (**bxPp**) We have highlighted the exact match evaluation metrics in Table 2 and Appendix E.
5. (**SrPL**) We have added one more potential limitation of CoK in Appendix G. CoK relies on the reasoning capability of LLMs, failure cases may stem from reasoning failures of LLMs. We also include three examples of these failure cases in Appendix H.2.
6. (**SrPL**) We have added more training details in Appendix D.6, including the base model, training epochs, batch size and other detailed settings. To ensure reproducibility, we will release our training code, data, and model weights


Thank you all once again for the valuable feedback and for facilitating the improvement of our work.

Best Regards,

Authors

---

### Meta-Review · Area_Chair_f6q1 · 2023-12-04

**Metareview:**

This paper proposes a novel framework named chain-of-knowledge (CoK), which augments large language models (LLMs) by dynamically incorporating grounding information from heterogeneous sources. An adaptive query generator (AQG) is introduced to access both unstructured and structured knowledge sources.

A reasonable amount of discussions took place between the authors and the reviewers. In the end, we got four reviews with ratings of 6, 6, 6, and 6 with confidence of 4, 4, 3, and 3 respectively.

The framework is considered novel (reviewers 84zL, bxPp, TzUb), and the experiments are comprehensive and well-conducted. (reviewers bxPp, SrPL, TzUb).  However, the issues raised by the reviews are critical, for instance, the detail of the framework (bxPp, SrPL, 84zL), the capability of proposed AQG (SrPL), contribution (SrPL, TzUb), and the experiment setting (TzUb). Fortunately, the authors have addressed the main concern proposed by the reviewers (84zL, bxPp, SrPL, TzUb).

**Justification For Why Not Higher Score:**

This paper is considered as good but some issues raised by the reviewed would need to be further addressed.

**Justification For Why Not Lower Score:**

Most concerns that were raised by reviewers have been well-addressed by the authors and the reviewers are consensus to be accepted with score at least 6.

---

### Decision · Program_Chairs · 2024-01-16

Accept (poster)